# Metabolomic Signature of Diabetic Kidney Disease in Cerebrospinal Fluid and Plasma of Patients with Type 2 Diabetes Using Liquid Chromatography-Mass Spectrometry

**DOI:** 10.3390/diagnostics12112626

**Published:** 2022-10-29

**Authors:** Huan-Tang Lin, Mei-Ling Cheng, Chi-Jen Lo, Gigin Lin, Fu-Chao Liu

**Affiliations:** 1Department of Anesthesiology, Chang Gung Memorial Hospital, College of Medicine, Chang Gung University, Taoyuan 333, Taiwan; 2Graduate Institute of Clinical Medical Sciences, College of Medicine, Chang Gung University, Taoyuan 333, Taiwan; 3Metabolomics Core Laboratory, Healthy Aging Research Center, Chang Gung University, Taoyuan 333, Taiwan; 4Department of Biomedical Sciences, College of Medicine, Chang Gung University, Taoyuan 333, Taiwan; 5Clinical Metabolomics Core Laboratory, Chang Gung Memorial Hospital, Taoyuan 333, Taiwan; 6Department of Medical Imaging and Intervention, Imaging Core Laboratory, Chang Gung Memorial Hospital, Taoyuan 333, Taiwan

**Keywords:** diabetes mellitus, cerebrospinal fluid, metabolomics, diabetic kidney disease, mass spectrometry

## Abstract

Diabetic kidney disease (DKD) is the major cause of end stage renal disease in patients with type 2 diabetes mellitus (T2DM). The subtle metabolic changes in plasma and cerebrospinal fluid (CSF) might precede the development of DKD by years. In this longitudinal study, CSF and plasma samples were collected from 28 patients with T2DM and 25 controls, during spinal anesthesia for elective surgery in 2017. These samples were analyzed using liquid chromatography-mass spectrometry (LC-MS) in 2017, and the results were correlated with current DKD in 2017, and the development of new-onset DKD, in 2021. Comparing patients with T2DM having new-onset DKD with those without DKD, revealed significantly increased CSF tryptophan and plasma uric acid levels, whereas phosphatidylcholine 36:4 was lower. The altered metabolites in the current DKD cases were uric acid and paraxanthine in the CSF and uric acid, L-acetylcarnitine, bilirubin, and phosphatidylethanolamine 38:4 in the plasma. These metabolic alterations suggest the defective mitochondrial fatty acid oxidation and purine and phospholipid metabolism in patients with DKD. A correlation analysis found CSF uric acid had an independent positive association with the urine albumin-to-creatinine ratio. In conclusion, these identified CSF and plasma biomarkers of DKD in diabetic patients, might be valuable for monitoring the DKD progression.

## 1. Introduction

Type 2 diabetes mellitus (T2DM) is characterized by chronic hyperglycemia, due to insulin resistance. T2DM is a pandemic that has affected more than 10.5% of the adult population (536.6 million) worldwide, in 2021, according to the International Diabetes Foundation’s Diabetes Atlas estimation [1]. Chronic hyperglycemia in patients with T2DM, results in significant long-term sequelae, including macrovascular and microvascular complications [2]. Diabetic kidney disease (DKD) occurs in 30–50% of the patients with T2DM and is now the leading cause of end-stage renal disease (ESRD), causing enormous healthcare and economic burden, worldwide [3]. DKD is a complex and heterogeneous disease that results in the glomerular inflammation and fibrosis [4]. The clinical diagnosis of DKD is based on the presentation of persistent albuminuria and the decreased estimated glomerular filtration rate (eGFR) [3]. Notably, DKD accounts for the majority of the excessive risk of cardiovascular and all-cause mortality in patients with T2DM [5]. Given the growing incidence of T2DM and the deleterious sequelae of DKD, reliable risk stratification of patients with T2DM, susceptible to develop DKD is required for an earlier intervention.

The kidneys are metabolically active during the modulation of the circulating metabolites and waste removal via filtration, reabsorption, and secretion. Chronic hyperglycemia and inflammation, shift the oxidative balance toward a pro-oxidative state, accelerating kidney damage and causing the DKD progression from the glomerular hyperfiltration to the declining glomerular filtration, the progressive albuminuria, and ultimately ESRD [6,7]. Hence, the metabolic and epigenetic dysregulation may be detected in blood and urine before the onset and diagnosis of DKD [8]. Metabolomic studies have been employed to investigate the biomarkers for DKD in blood and urine to gain in-depth insights into the pathophysiology of DKD, which would enable further biomarker research for potential therapeutic targets [8]. Liquid chromatography-mass spectrometry (LC-MS) is a high-throughput analytical platform, used to identify and quantify small metabolites. LC-MS is especially useful for the risk stratification and monitoring the disease severity to enhance the diagnosis and therapeutic efficacy [9]. An untargeted gas chromatography-mass spectrometry analysis of the urine metabolomics among patients with T2DM with or without DKD, found a disturbed mitochondrial and fatty acid metabolism in patients with DKD, which was confirmed by the reduced mitochondrial biogenesis and fatty acid oxidation in the renal histological examinations [9]. Serum and urine metabolomics of 286 European patients, revealed that the combination of plasma C-glycosyl tryptophan, pseudouridine, and N-acetylthreonine, was associated with a decline in renal function [10]. Another serum lipidomic LC-MS analysis found that a combination of lysophosphatidylethanolamine (LysoPE) (16:0), phosphatidylethanolamine (PE) (16:0/20:2), and triacylglycerol exhibited an excellent performance in the diagnosis of DKD [11].

Cerebrospinal fluid (CSF) exchanges systemic metabolites with the central nervous system (CNS), via a limited transcellular transport constituted by the specialized tight junction of the blood–brain barrier (BBB). The impaired BBB integrity in patients with T2DM has been recognized in MRI studies of human cohorts, which is ascribed to the tight junction disruption and enhanced paracellular permeability, due to chronic hyperglycemia [12]. Thus, the exaggerated metabolic derangement in patients with DKD, might induce subtle metabolomic changes through the BBB leakage, which could be detected in CSF, using sensitive mass spectrometry. Our group focuses on investigating the CSF metabolomic signature using CSF samples obtained during routine spinal anesthesia, and we found the decreased mitochondrial phosphorylation and increased anaerobic glycolysis in the CSF metabolomics of patients with T2DM, using NMR techniques [13]. During further correlating of the CSF metabolomic signature with diabetic complications, we found patients with DKD had a specific CSF metabolomic presentation which has not been well explored. To profile the metabolic alterations in diabetic patients who develop DKD, we conducted this longitudinal follow-up study of CSF and plasma samples from diabetic patients with DKD, in comparison with diabetic patients without DKD and control participants, using sensitive LC-MS analysis.

## 2. Materials and Methods

In this study, we explored the metabolomic signatures of current DKD and new-onset DKD, in the CSF and plasma samples of patients with T2DM, using LC-MS metabolomics. The study protocol was registered in the ClinicalTrials.gov database (Identifier: NCT03725709) and the study was approved by the Ethical Review Board of Chang Gung Medical Foundation, Taiwan (approval number: 201600122A3). The study protocol was clearly explained to each participant and informed consent was obtained before enrollment.

### 2.1. Study Population

This longitudinal follow-up study included 60 participants aged 20–70 years. The sample collection was performed between 1 January 2017 and 31 July 2017, while the participants underwent spinal anesthesia for elective surgeries, at Linkou Chang Gung Memorial Hospital, Taiwan. Thirty participants had a history of T2DM, while the other 30 control participants did not. The patients with a history of T2DM have a fasting glucose level of >126 mg/dL (classified as diabetes, according to the diagnostic criteria) and were taking oral hypoglycemic agents. The other 30 participants had a fasting blood glucose <126 mg/dL and formed the control group [14]. The CSF and blood samples of all 60 participants were collected and subjected to the LC-MS metabolomic analysis on 10 November 2017. Among the T2DM patients, eight patients fulfilled the diagnostic criteria of DKD with persistent albuminuria (spot urine albumin-to-creatinine ratio (UACR) > 30 mg/g) or a decreased eGFR <60 mL/min/1.73 m^2^ within 3 months before the sampling [15]. These eight participants were classified into the current DKD group. The 4-year longitudinal follow-up period extended until 31 December 2021. During this period, seven patients were lost to follow-up, and another eight patients who originally did not have DKD in 2017, developed DKD (new-onset DKD group) in 2021. The remaining 12 patients with T2DM had no evidence of DKD during the entire study period (without DKD group). The final cohort in this study included 25 control participants, eight patients with current DKD, eight patients with new-onset DKD, and 12 patients with T2DM but without DKD. The patient demographics including age, sex, height, weight, and chronic diseases, were recorded. Renal function evaluations, including the plasma creatinine level, eGFR, and spot UACR, were obtained from the latest laboratory results, prior to the CSF and blood sampling.

### 2.2. CSF and Blood Sampling Procedures

The sample collection procedures were largely the same as those used in our previous NMR metabolomic study of patients with T2DM [13]. In this study, the registered anesthesiologists H.T. Lin and F.C. Liu, applied the CSF sampling and spinal anesthesia for these enrolled participants. Once they obtained the informed consent, the enrolled participants received optional spinal anesthesia with a 26-gauge spinal needle at the L3 and L4 interspace. Following the successful outflow of clear CSF, 1.2 mL of the CSF was collected into a polypropylene tube. In addition, 4 mL of blood was collected in an EDTA-coated tube 10 min before the CSF collection. Plasma was collected by centrifuging the blood samples for 5 min at 10,000 rpm at 4 °C. Then, 500 μL of the plasma and CSF samples were sent for biochemical analyses. The rest of the CSF and plasma samples were aliquoted and stored at −80 °C for subsequent LC-MS metabolomic analyses. None of the participants reported discomfort during the CSF and blood sample collection procedures.

### 2.3. Biochemical Analyses of the CSF and Plasma Samples

The biochemical parameter analyzed in the 500 μL of plasma and CSF samples, included plasma glucose, plasma glycated hemoglobin A1c (HbA1c), plasma insulin, CSF glucose, and CSF insulin levels. The plasma and CSF glucose levels were determined using the glucose oxidase assay provided by Cell Biolabs, San Diego, CA, USA. Insulin and HbA1c levels were quantified using enzyme-linked immunosorbent assay (ELISA) kits from Mercodia (Uppsala, Sweden) and Cloud-Clone Corporation (Houston, TX, USA), respectively. The degree of insulin resistance in patients with T2DM was compared using the homeostasis model assessment of the insulin resistance index (HOMA-IR), calculated as fasting glucose (mg/dL) × fasting insulin (mU/L) divided by 405 [16].

### 2.4. Sample Preparation for the LC-MS

The CSF samples (50 μL) were mixed with cooled acetonitrile (400 μL) and 0.1% formic acid to precipitate proteins. Following the centrifugation at 12,000× *g* for 30 min, the supernatant was transferred. The pellet was resuspended in 400 μL of 50% methanol with 0.1% formic acid. Following the centrifugation at 12,000× *g* for 30 min, the two supernatants were mixed and dried under nitrogen gas. The residue was suspended in 100 μL of 50% acetonitrile with 0.1% formic acid for LC-MS analysis.

The liquid chromatographic separation was conducted on an Acquity UPLC BEH C8 column (1.7 μm, 2.1 × 100 mm^2^; Waters, Milford, MA, USA) using an Acquity TM Ultra Performance Liquid Chromatography (UPLC) system (Waters Corporation). The column was maintained at 45 °C and the flow rate was 0.5 mL/min. The mobile phase consisted of 0.1% formic acid in water (Phase A) and acetonitrile, containing 0.1% formic acid (Phase B). The mass spectrometry was performed using a Waters Q TOF-MS (SYNAPT G2S; Waters MS Technologies, Manchester, UK) operated in ESI positive and negative ion modes. The scan range was 20–990 *m*/*z*. The desolvation gas flow rate was 700 L/h at 300 °C. The source-cone voltage was set to 35 V. The capillary voltage were 2.7 kV in the positive mode and 2 kV in the negative mode. The lock mass was leucine encephalin (*m*/*z*: 120.0813 and 556.2771 for the positive mode and *m*/*z*: 236.1035 and 554.2615 for the negative mode). The LC-MS multivariate data was statistically analyzed with the mean centering and the Pareto scaling using soft independent modeling of class analogy software (SIMCA-P+, version 13.0; Umetrics, Umea, Sweden).

### 2.5. Metabolites Identification and the Statistical Analysis

The metabolite identification was performed by comparing the chemical shifts and multiplicity patterns of each metabolite with the Human Metabolome Database (HMDB) [17]. The identified metabolites were analyzed and compared, using the fold change, Akaike information criterion (AIC), the area under the receiver operating characteristic curve (AUC) value, and the odds ratio (OR). Other metabolomic analyses, such as the enrichment analysis, were performed using the online tool MetaboAnalyst 5.0 [18].

These collected data are presented as means ± SD for the continuous variables (such as body weight and the serum creatinine levels) and as a percentage for the qualitative variables (such as sex and chronic diseases). The statistical analyses were based on the orthogonal partial least-squares discriminant analysis (OPLS-DA) coefficients in the LC-MS signals. Comparisons between the two groups were performed using the student’s *t*-test or chi-squared test and the analysis of variance (ANOVA), for the comparisons among multiple groups. In this study, correlating the CSF and plasma biomarkers with current DKD in 2017, was our expected primary outcome, and correlating the biomarkers with new-onset DKD in 2021, was the expected secondary outcome. First, we identified significant metabolites in the CSF and plasma samples that could discriminate between patients with DKD, without DKD, and the control participants. We then selected significant metabolites in the CSF and plasma samples to construct the metabolite combinations. We selected the target panels with the highest AUC values as the final result. Since diabetic patients had a significantly higher age, BMI, male percentage, and chronic diseases, such as hypertension and hyperlipidemia, compared with that of the control participants, these confounding factors were adjusted in further comparisons. In addition, the correlation of these metabolites with standard renal function measurements, such as UACR and eGFR, was calculated using a regression model to compare their association. All statistical analyses were executed using the SAS software (version 9.4; SAS Institute Inc. Cary, NC, USA), and a two-sided *p* value < 0.05 was dictated as statistically significant.

## 3. Results

### 3.1. Group Separation and Their Demographic Comparison

Our final cohort included 28 patients with T2DM (12 with no DKD, eight had current DKD in 2017, and eight had new-onset DKD in 2021) and 25 control participants. The analysis protocol is illustrated in Figure 1. A comparison of the demographic and biochemical parameters of the enrolled participants is shown in Table 1. The demographic comparison showed that the diabetic patients had a significantly higher male percentage, BMI, hypertension, hyperlipidemia, CSF glucose, plasma glucose, plasma HbA1c, and HOMA-IR, compared with that of the control participants. Owing to these significant demographic differences, the subsequent analyses were adjusted for sex, BMI, age, and chronic diseases (hypertension and hyperlipidemia). Three patients with current DKD in 2017, had a deteriorated renal function, requiring dialysis in 2021. In addition, T2DM patients with DKD had more concomitant diabetic retinopathy or neuropathy, than the T2DM patients without DKD.

### 3.2. OPLS-DA Score Plots

The OPLS-DA score plots of the LC-MS signal integrations in the CSF and plasma samples from patients with DKD, versus the control participants, are compared in Figure 2. The OPLS-DA score plots showed a clear discrimination between the diabetic patients and the control participants and a superior discrimination in the plasma samples, compared with the CSF samples. Appendix A compares the OPLS-DA score plots of the patients with T2DM and DKD with that of the patients without DKD. The discrimination between the patients with DKD and the patients without DKD was less significant than the discrimination between the diabetic patients and the control participants.

### 3.3. Metabolomic Comparison between the Patients with DKD versus the Control Participants

The comparison between the altered metabolites in the CSF samples from the diabetic patients with current or new-onset DKD and the patients without DKD or the control participants is shown in Table 2, and the comparison for the plasma samples is shown in Table 3. The LC-MS signal integration in the patients with current DKD showed significantly higher uric acid and lower paraxanthine levels in the CSF samples than in the patients without DKD and the control participants (adjusted fold change >1.2 or <0.8, and *p* < 0.05). The plasma samples from patients with current DKD had significantly higher L-acetylcarnitine and uric acid and lower phosphatidylethanolamine (PE) 38:4, phosphatidylcholine (PC) 36:4, and bilirubin levels than in the samples from patients without DKD. As for the new-onset DKD, these patients had significantly higher CSF tryptophan levels than the patients without DKD. The plasma samples of patients with new-onset DKD, showed significantly higher levels of uric acid and lower levels of lysophosphatidylcholine (LysoPC) 18:2, PC 38:6, and PC 36:4.

### 3.4. Metabolite Combinations for Correlating with DKD

To identify the biomarkers correlating with our DKD outcomes, we constructed metabolite combinations using CSF and plasma metabolites with a significant discrimination between patients with or without DKD. We then compared the AIC values, AUC values, and adjusted the ORs for discriminating the current DKD or new-onset DKD, from the patients without DKD and the control participants, using ANOVA and a multivariate analysis (CSF metabolites in Table 4 and plasma metabolites in Table 5). Among these metabolite combinations for the discriminating of current DKD, a CSF combination of uric acid and paraxanthine (AUC: 0.748 in comparison with current DKD vs. without DKD), and a plasma combination of L-acetylcarnitine, bilirubin, uric acid, and PC 36:4 (AUC: 0.897 in comparing current DKD vs. without DKD) had comparatively lower AIC values, significant AUC values, and adjusted ORs. In contrast, the CSF metabolite of tryptophan (AUC: 0.745 in comparing new-onset DKD vs. without DKD) and plasma combination of uric acid, PC 38:6, and PC 36:4 (AUC: 0.817 in comparing new-onset DKD vs. without DKD) had relatively higher AUC values for discriminating the new-onset DKD from the patients without DKD. Figure 3 depicts the AUC curve of these metabolite combinations for the correlation with the current DKD or the new-onset DKD in the CSF and plasma samples.

### 3.5. Correlation Analysis of the Altered Metabolites with UACR and eGFR

We also performed a correlation analysis to compare the association of these significantly altered metabolites in patients with DKD with standard renal function measurements, such as UACR and eGFR. Table 6 presents the results of the correlation analysis. Among the CSF metabolites, uric acid, and hypoxanthine were independently positively correlated with UACR and negatively correlated with eGFR, whereas paraxanthine had a negative correlation with UACR and a positive correlation with eGFR. Among the significantly altered plasma metabolites in patients with DKD, uric acid and L-acetylcarnitine were independently positively correlated with UACR but negatively correlated with eGFR, whereas PC 36:4 and PE 38:4 were negatively correlated with UACR and positively correlated with eGFR.

### 3.6. Enrichment Analysis and Metabolic Pathways of the Altered Metabolites in DKD

The enrichment analysis of the altered metabolites for the current DKD showed that caffeine, glycerophospholipid, and purine metabolism were affected. In contrast, purine, glycerophospholipid, tryptophan metabolism, and the aminoacyl-tRNA biosynthesis were involved in the altered pathways of the new-onset DKD (Figure 4). Regarding the metabolites profiled in the pathogenesis of DKD, the altered metabolic pathways during the DKD development are depicted in Figure 5. Altogether, the profiled metabolites may imply a defective mitochondrial fatty acid oxidation, phospholipid remodeling, excessive oxidative stress, and altered purine metabolism in patients with DKD; and exaggerated metabolic changes in the systemic circulation may be present in the CSF samples through the BBB leakage in patients with T2DM.

## 4. Discussion

This longitudinal follow-up cohort study aimed to identify the CSF and plasma metabolomic signatures correlated with current DKD and new-onset DKD, in patients with T2DM using LC-MS. The metabolomic analysis showed significantly altered levels of uric acid and paraxanthine in the CSF samples, whereas that of L-acetylcarnitine, uric acid, bilirubin, and PC 36:4 were altered in the plasma samples from patients with current DKD. In addition, significantly higher CSF tryptophan and altered plasma uric acid, PC 38:6, and PC 36:4 levels were associated with the development of new-onset DKD in patients with T2DM. Uric acid showed an independently positive correlation with UACR and a negative correlation with eGFR in both the CSF and plasma samples from patients with DKD. The profiled CSF and plasma metabolome of patients with DKD revealed defective mitochondrial fatty acid, tryptophan, and purine metabolism in patients with T2DM complicated by DKD. In addition, uric acid was found to be significantly elevated in both plasma and CSF samples of patients with DKD, implying its importance in the pathophysiology of the DKD progression.

The two most common metabolomic analytical methods are NMR spectroscopy and MS. NMR spectroscopy can identify core metabolites in key metabolic pathways, whereas MS can identify low- abundance metabolites with a wide detection range, an excellent sensitivity, and precise quantification capabilities [19]. In our previous CSF and plasma metabolomic profiling of patients with T2DM, using NMR spectrometry, we found that a panel of CSF alanine, histidine, leucine, pyruvate, tyrosine, and valine correlated well with the presence of T2DM, which suggests a mitochondrial dysfunction in the cerebral circulation of these patients [13]. In the current LC-MS analysis of patients with DKD, we identified specific DKD-correlated metabolites, including uric acid, acetylcarnitine, bilirubin, and phospholipids, providing a deeper insight into the metabolomic changes in DKD. These profiled metabolites are involved in the cell membrane turnover, redox reactions, neurotransmitter metabolism, and mitochondrial respiration, and were not identified in our previous NMR metabolomic analysis. Combining the results of the previous NMR and the current LC-MS metabolomic analyses, we can gain a deeper understanding of the glucose hypometabolism, BBB breakdown, neuroinflammation, and mitochondrial dysfunction in patients with T2DM complicated with DKD.

Uric acid, a natural scavenger of peroxynitrite, is an intermediate product of the purine metabolism, and its concentration reflects antioxidant activity [20]. Changes in blood and CSF uric acid levels have been reported in patients with multiple sclerosis and Guillain–Barré syndrome [20,21]. The high CSF uric acid level in these neurological diseases might be correlated with an impaired BBB in these patients [20]. Increased serum uric acid levels have been associated with a higher risk of DKD, in patients with type 1 diabetes, T2DM, and chronic kidney disease, possibly through their contribution to tubular fibrosis [22,23]. A large cross-sectional study of patients with T2DM found a positive correlation between the serum uric acid level and albuminuria and a negative correlation with eGFR after adjusting for the confounding factors [24]. Since serum uric acid is a recognized biomarker of DKD, our plausible finding that CSF uric acid might be a possible biomarker of new-onset DKD, deserves further verification [25].

Tryptophan, an essential amino acid, is metabolized mainly by the indole (<5%) and kynureine (95%) pathway [8]. Downstream metabolites of tryptophan, such as kynurenic acid and NAD, contribute to the enhanced oxidative stress and inflammation in endothelial cells, leading to the development and progression of DKD [10,26]. In the previous serum metabolomic analyses, tryptophan was regarded as a potential prognostic marker for DKD, and the increased serum levels of tryptophan (or tryptophan/kynurenine ratio) were inversely associated with renal function deterioration in patients with DKD [8,27]. Our finding of the increased CSF tryptophan in new-onset DKD but similar levels in patients with current DKD and without DKD, might be explained by the initial BBB leakage of tryptophan in patients with new-onset DKD, followed by the tryptophan downregulation during the DKD progression. However, this hypothesis requires further quantification of the downstream tryptophan metabolites, which is beyond the scope of the current experimental design.

Caffeine, the most extensively consumed psychoactive drug with adenosine antagonist properties, has neuroprotective effects in some neurological disorders, such as Parkinson’s disease (PD) [28]. A large cohort study comparing 368 individuals with PD and unaffected control participants found lower plasma and CSF levels of caffeine and its downstream metabolites (paraxanthine) in patients with PD [28]. In patients with traumatic brain injury, increased CSF concentrations of caffeine and paraxanthine were associated with favorable outcomes [29]. The possible mechanism underlying this phenomenon might be explained by the neuroprotective effect of caffeine through the long-term upregulation of the adenosine A1 receptors [29]. Therefore, our finding that lower CSF paraxanthine levels in patients with current DKD may imply the loss of the neuroprotective effects of caffeine in patients with DKD; however, further verification is required.

Acylcarnitine is generated by the fatty acid oxidation inside the mitochondria, and its accumulation is associated with insulin resistance and mitochondrial dysfunction [30]. L-Acetylcarnitine is the shortest and most common (75%) form of acylcarnitine. The carnitine shuttle system imports long-chain fatty acids into the mitochondria for oxidation, and the conversion of acylcarnitine into acyl-CoA and free carnitine, makes acyl-CoA available for further production of acetyl-CoA (through β-oxidation), which enters the tricarboxylic acid cycle for energy production [31]. In diabetic patients, incomplete fatty acid oxidation and lipotoxicity contribute to increased serum acylcarnitine levels [31]. The lipidomic analyses of patients with DKD revealed that the concentration of acylcarnitine of different lengths varied in different DKD stages [32]. In the early stage of DKD with normoalbuminuria or microalbuminuria, the adaptive compensation for fatty acid oxidation increases the long-chain acylcarnitine [32]. However, in advanced DKD with macroalbuminuria, the incomplete fatty acids oxidation of long-chain fatty acids contributes to an increase in the short- and medium-chain acylcarnitine [32]. Thus, acylcarnitines may serve as sensitive biomarkers for the risk stratification and staging of DKD [33]. In our study, the accumulation of acetylcarnitine in the plasma of patients with current DKD may imply an impaired mitochondrial function and fatty acid oxidation in patients with DKD [32].

Bilirubin, a powerful endogenous antioxidant with anti-inflammatory properties, can act on cellular pathways to halt the progression of DKD [6]. Owing to the protective effect of bilirubin in DKD by counteracting the oxidative stress and fibrosis, recent experimental and clinical studies have shown that an increase in serum bilirubin concentrations slowed down the progression of DKD [6,34] Based on our analysis and previous studies, the lower plasma bilirubin level in our patients with current DKD may imply the protective role of bilirubin in DKD.

Altered lipid levels, including sphingolipids and phosphatidylcholines (PCs), are associated with renal function impairment in diabetic patients [23]. Sphingolipids (including sphingomyelin and ceramide) are important constituents of the cell membrane and are involved in cell signalling and activation. Sphingolipid accumulation in renal glomeruli has been hypothesized to be a major contributor to the glomerular proliferation and kidney fibrosis in DKD [23]. PEs and PCs are two of the most abundant phospholipids in mammalian cells, comprising 15–25% and 40–50% of the total cellular phospholipids, respectively. The PE/PC ratio often increases in patients with DKD, and the subsequent decrease in the cell membrane fluidity increases the cell membrane permeability, further contributing to cellular damage [11]. PCs undergo hydrolysis on their acyl fatty acid chains by phospholipases to form LysoPCs and arachidonic acid, which contribute to the downstream production of leukotrienes and the inflammatory response [35]. In a large plasma LC-MS metabolomic study of patients with DKD, the accumulation of acylcarnitine, the reduction of PCs, and the elevation of long-chain sphingomyelin and ceramide levels in these patients, suggested a phospholipid remodeling in DKD [15]. Thus, the decrease of PE 38:4, PC 36:4, and PC 38:6 levels in patients with DKD, may suggest lipotoxicity and phospholipid remodeling in patients with DKD. In addition, the increased ratio of PE/PC and LysoPC/PC in our patients with DKD, may also suggest an increased membrane permeability and inflammatory response in DKD.

Altogether, the increased uric acid and lower paraxanthine levels in the fasting CSF samples of DKD patients, might suggest an enhanced oxidative stress and decreased neuroprotective effects of caffeine. The altered uric acid, acetylcarnitine, bilirubin, and PCs in the plasma samples of patients with DKD may imply a mitochondrial dysfunction and phospholipid remodeling.

Theoretically, better glycemic and blood pressure control might be able to ameliorate the progression of DKD, but it is still unable to substantially decrease the annual incidence of DKD-related ESRD [36]. Thus, a novel predictor of the high risk of developing DKD in diabetic patients is needed. Metabolomics is a powerful approach that enables us to explore the molecular pathophysiology and improve the clinical management of DKD and its complications. Our profiled DKD biomarkers, including the altered uric acid, paraxanthine, and tryptophan levels in CSF and the decreased plasma phospholipids, can be used for the risk stratification of patients with T2DM more susceptible to developing DKD to achieve an early intervention.

To the best of our knowledge, this pilot study is the first to predict the development of DKD by integrating CSF and plasma metabolomic LC-MS signatures of Taiwanese patients with T2DM. Although the invasive nature of the CSF sampling may limit the utilization of CSF biomarkers, these profiled CSF and plasma metabolites could deepen our insight into the DKD pathophysiology. The method of CSF sampling undertaken in this study enabled us to detect the actual metabolite alterations in the cerebral circulation without causing unnecessary discomfort. The diagnosis of DKD in this cohort was solid because it was confirmed by specialists during inpatient or out-patient follow-ups for T2DM.

However, this small cohort study had several limitations. First, the number of cases in this study were limited due to insufficient budget and loss to follow-up during the COVID-19 pandemic. Therefore, the interpretation of the current results should be made with caution because of individual variations. However, our novel findings regarding the DKD-correlated CSF and plasma metabolites deserve further validation in larger cohorts. Second, the enrolled participants had significant variations among different groups in sex, BMI, serum creatinine, and chronic diseases (hypertension and hyperlipidemia), which might confound our results, even though these factors were already adjusted. Moreover, the complex and dynamic nature of the metabolome could be affected by many factors, such as socioeconomic circumstances, diet, lifestyle, and medications. Further larger cohorts and longitudinal repeated samplings might be required to verify our findings and determine the causal relationship.

## 5. Conclusions

In this longitudinal study, we profiled the CSF and plasma metabolomic signature of patients with T2DM complicated with current DKD or developing new-onset DKD, using the LC-MS analysis. The identified potential DKD biomarkers, such as uric acid, acetylcarnitine, PC 36:4, and PE 38:4, had a significant correlation with eGFR and UACR, suggesting their values in the predictions and risk stratification for DKD in patients with T2DM. These metabolic alterations imply a defective mitochondrial fatty acid oxidation, purine metabolism, and phospholipid remodeling during the DKD progression. Further verification of our results in a larger multi-omic cohort is required to confirm the causal relationship.

## Figures and Tables

**Figure 1 diagnostics-12-02626-f001:**
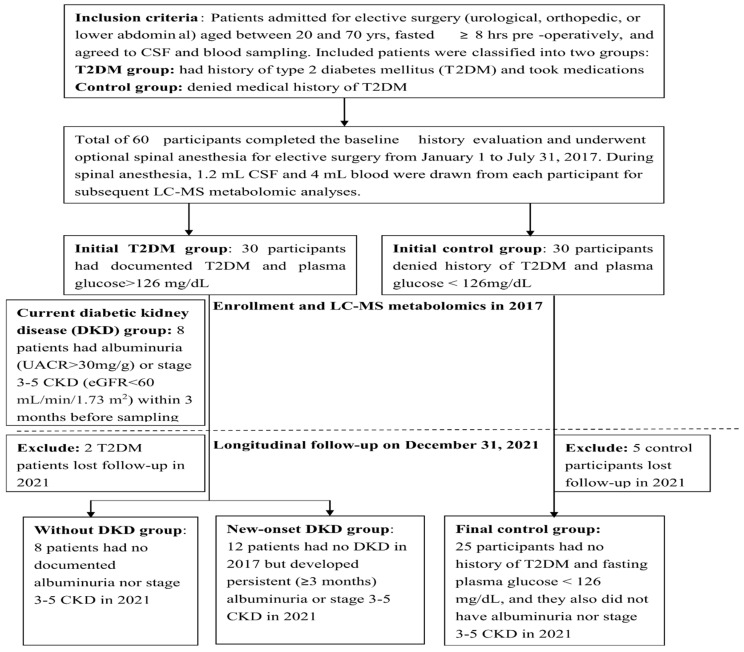
Flow chart for the study design and group separation.

**Figure 2 diagnostics-12-02626-f002:**
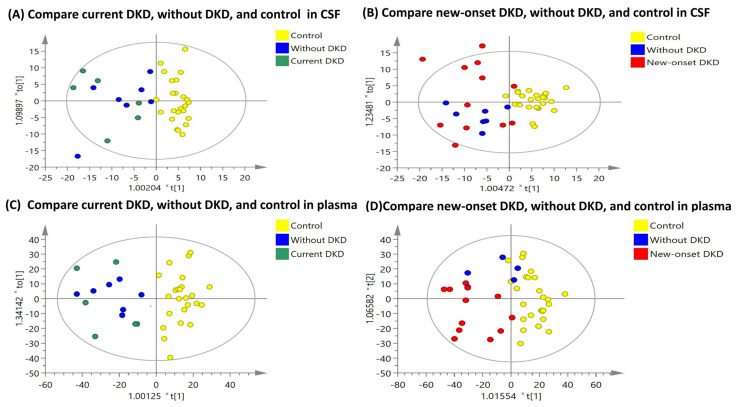
Orthogonal partial least-squares discriminant analysis (OPLS-DA) score plots in (**A**) CSF samples for the comparison among the diabetic patients with current DKD, diabetic patients without DKD, and the control participants (reliability: R^2^X = 0.516, R^2^Y = 0.450, Q^2^ = 0.275); (**B**) CSF samples for the comparison among diabetic patients with new-onset DKD, diabetic patients without DKD, and the control participants (reliability: R^2^X = 0.428, R^2^Y = 0.461, Q^2^ = 0.311); (**C**) plasma samples for the comparison among the diabetic patients with current DKD, the diabetic patients without DKD, and the control participants (reliability: R^2^X = 0.550, R^2^Y = 0.498, Q^2^ = 0.304); (**D**) plasma samples for the comparison among the diabetic patients with new-onset DKD, the diabetic patients without DKD, and the control participants (reliability: R^2^X = 0.530, R^2^Y = 0.514, Q^2^ = 0.413).

**Figure 3 diagnostics-12-02626-f003:**
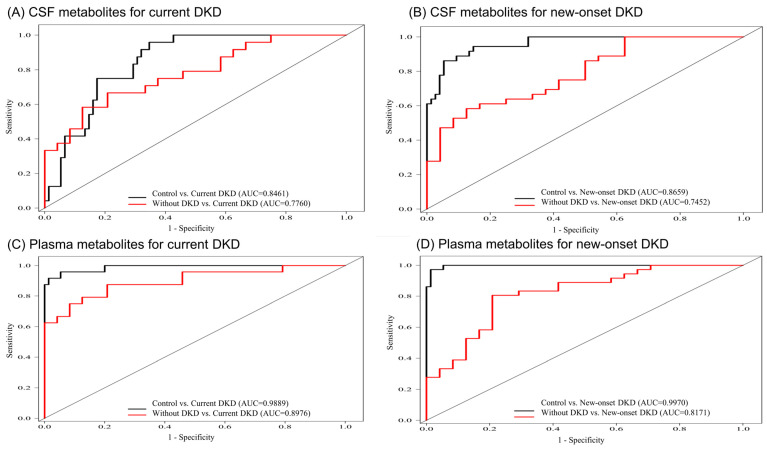
Area under curve of the significant altered metabolite combinations in the CSF samples for discriminating patients with (**A**) current DKD; (**B**) new-onset DKD; and metabolite combinations in the plasma samples for discriminating patients with (**C**) current DKD; (**D**) new-onset DKD, from patients without DKD and the control participants.

**Figure 4 diagnostics-12-02626-f004:**
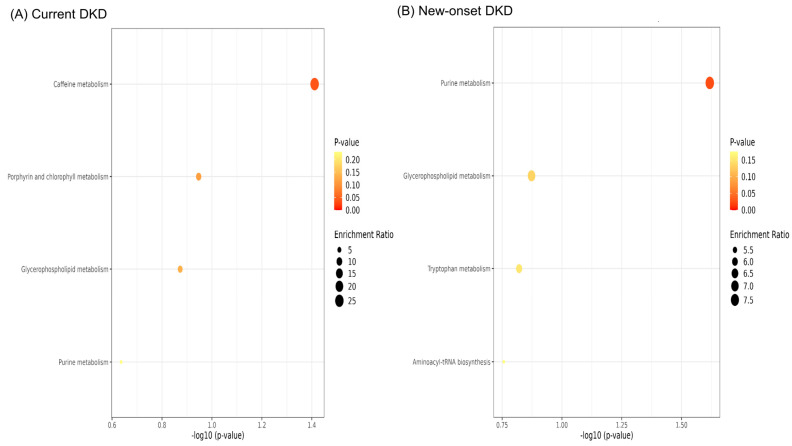
Enrichment analysis of the involved metabolic pathways for significant metabolites in (**A**) current diabetic kidney disease and (**B**) new-onset diabetic kidney disease.

**Figure 5 diagnostics-12-02626-f005:**
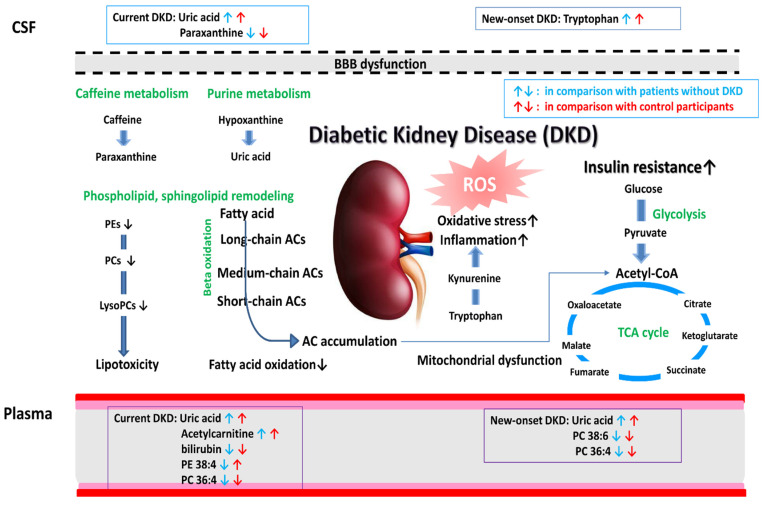
Schematic representation of the significant metabolite alterations in diabetic kidney disease.

**Table 1 diagnostics-12-02626-t001:** Demographic comparison between the type 2 diabetic patients with or without diabetic kidney disease and the control participants.

Variables	Control (*n* = 25)	Without DKD (*n* = 8)	Current DKD (*n* = 8) ^c^	New-Onset DKD (*n* = 12) ^c^	*p*
Male sex, N (%)	11 (45.83%)	7 (87.50%)	5 (62.50%)	10 (83.33%)	0.010 *
Age (mean ± SD, years)	50.72 ± 14.67	59.37 ± 8.75	51.25 ± 10.49	60.33 ± 8.95	0.086
BMI (kg/m^2^)	21.99 ± 3.49	26.89 ± 4.19	27.86 ± 5.66	24.55 ± 3.79	0.002 *
DM duration (years)	NA	4.87 ± 2.10	10.62 ± 4.59	11.08 ± 8.08	0.074
**Current medications, N (%)**					
Insulin injection	NA	1 (12.50%)	6 (75.00%)	7 (58.33%)	0.030 *
Anti-hypertensive agents	1 (4.16%)	4 (50.00%)	2 (25.00%)	5 (41.66%)	0.012 *
Lipid-modifying agents	1 (4.16%)	2 (25.00%)	3 (37.50%)	5 (41.66%)	0.028 *
**Biochemical parameters**					
CSF glucose (mg/dL)	55.23 ± 21.12	85.90 ± 32.57	79.79 ± 31.95	78.79 ± 22.59	0.006 *
CSF insulin (mU/L)	0.21 ± 0.10	0.36 ± 0.21	0.21 ± 0.18	0.26 ± 0.14	0.195
Plasma glucose (mg/dL)	93.07 ± 18.37	181.68 ± 69.21	158.89 ± 84.33	160.43 ± 73.11	<0.001 *
Plasma HbA1c (%) ^a^	5.73 ± 0.32	10.08 ± 2.62	11.46 ± 2.76	9.95 ± 2.31	0.017 *
Plasma insulin (mU/L)	7.25 ± 4.78	12.65 ± 5.96	7.20 ± 5.07	7.88 ± 2.72	0.044 *
Plasma HOMA-IR ^b^	2.03 ± 1.65	5.66 ± 4.09	3.46 ± 4.65	3.18 ± 1.76	0.024 *
**Renal function in 2017**					
Serum creatinine (mg/dL)	0.78 ± 0.26	1.02 ± 0.28	1.62 ± 1.04	1.08 ± 0.46	0.002 *
eGFR (ml/min per1.73 m^2^)	109.04 ± 25.56	93.75 ± 24.78	78.12 ± 35.39	98.00 ± 17.85	0.033 *
UACR	NA	14.24 ± 10.06	33.02 ± 18.86	2191.75 ± 1419.63	0.002 *
Dialysis in 2017	NA	0 (0%)	0 (0%)	0 (0%)	1.000
**Renal function in 2021**					
Serum creatinine (mg/dL)	0.81 ± 0.18	0.82 ± 0.31	6.20 ± 4.26	1.69 ± 1.03	<0.001 *
eGFR (ml/min per1.73 m^2^)	95.64 ± 24.15	125.75 ± 85.18	26.63 ± 34.84	57.92 ± 27.70	<0.001 *
UACR	NA	15.30 ± 12.87	5480.87± 2706.05	1420.00 ± 1234.02	<0.001 *
Dialysis in 2021	NA	0 (0%)	3 (37.50%)	0 (0%)	0.034 *
**Diabetic retinopathy**	NA	2 (25.00%)	7 (87.50%)	9 (75.00%)	0.039 *
**Diabetic neuropathy**	NA	0 (0%)	3 (37.50%)	5 (41.66%)	0.111

Abbreviation: T2DM, type 2 diabetes mellitus; DKD, diabetic kidney disease; BMI, body mass index; HOMA-IR, homeostatic model assessment for insulin resistance; HbA1c, glycated hemoglobin A1c; eGFR, estimated glomerular filtration rate; UACR, urinary albumin-to-creatinine ratio. * *p* < 0.05. ^a^ HbA1c was expressed in DCCT/NGSP unit (%); it would be 82 ± 26 in T2DM and 36 ± 18 in control in IFCC unit (mmol/mol). ^b^ HOMA-IR = glucose (mg/dL) × insulin (mU/L)/405. ^c^ diabetic kidney disease definition is the presence of persistent (≥3 months) albuminuria (spot UACR > 30 mg/g) or existence of stage 3–5 chronic kidney disease (eGFR < 60 mL/min/1.73 m^2^).

**Table 2 diagnostics-12-02626-t002:** Altered metabolites in the CSF samples of the type 2 diabetic patients with DKD, compared to patients without DKD or the control participants.

Metabolites in CSF	LC-MS Signal Integration (Mean ± SD) (×10^−3^ a.u.)				Adjusted Fold Change ^a,#^			
	Control	Without DKD	Current DKD	New-Onset DKD	Compared with Control		Compared with without DKD	
					Current DKD	New-Onset DKD	Current DKD	New-Onset DKD
Proline betaine	10.12 ± 4.48	61.74 ± 8.85	62.16 ± 8.13	52.17 ± 6.57	5.139 *	5.155 *	1.035	0.845
Tryptophan	36.12 ± 5.16	54.63 ± 10.18	52.16 ± 7.07	112.74 ± 7.56	1.423 *	3.121 *	0.968	2.063 *
D-glucose	83.76 ± 1.94	98.05 ± 3.82	103.93 ± 3.94	95.99 ± 2.83	1.245 *	1.145 *	1.066	0.979
Phenylalanine	135.66 ± 3.65	143.32 ± 7.21	132.82 ± 6.35	147.24 ± 5.35	0.954	1.085	0.997	1.027
Uric acid	52.31 ± 2.19	46.67 ± 4.32	72.03 ± 4.10	55.63 ± 3.21	1.363 *	1.063	1.523 *	1.192
L-acetylcarnitine	29.96 ± 1.37	31.98 ± 2.69	37.65 ± 2.33	28.57 ± 2.00	1.208 *	0.953	1.192	0.893
Paraxanthine	19.89 ± 2.87	14.89 ± 5.67	2.43 ± 5.19	17.32 ± 4.21	0.056 *	0.871	0.074 *	1.163
Hypoxanthine	30.79 ± 0.87	31.28 ± 1.72	29.64 ± 1.46	25.79 ± 1.27	0.985	0.837 *	0.916	0.825 *
Creatinine	14.34 ± 0.32	13.81 ± 0.64	13.67 ± 0.61	11.77 ± 0.47	0.957	0.821 *	1.015	0.852 *

Abbreviation: T2DM, type 2 diabetes mellitus; DKD, diabetic kidney disease. ^a^ fold change was adjusted for age, sex, body mass index (BMI), and chronic diseases (hypertension and hyperlipidemia). ^#^ Fold change and *p* value were calculated using two sample *t* tests. * *p* < 0.05.

**Table 3 diagnostics-12-02626-t003:** Altered metabolites in the plasma samples of the type 2 diabetic patients with DKD, compared to the patients without DKD or the control participants.

Metabolites in Plasma	LC-MS Signal Integration (Mean ± SD) (×10^−3^ a.u.)				Adjusted Fold Change ^a,#^			
	Control	Without DKD	Current DKD	New-Onset DKD	Compared with Control		Compared with without DKD	
					Current DKD	New-Onset DKD	Current DKD	New-Onset DKD
Proline betaine	9.02 ± 4.46	53.74 ± 8.79	62.97 ± 7.39	50.93 ± 6.53	6.187 *	5.646 *	1.285	0.947
Uric acid	10.92 ± 0.68	14.85 ± 1.35	24.67 ± 1.08	20.95 ± 1.00	2.253 *	1.918 *	1.668 *	1.411 *
D-glucose	37.27 ± 1.38	59.35 ± 2.74	61.27 ± 2.78	65.02 ± 2.03	1.664 *	1.744 *	1.049	1.096
L-acetylcarnitine	28.16 ± 1.29	37.84 ± 2.55	47.81 ± 2.38	39.36 ± 1.89	1.649 *	1.398 *	1.163 *	1.042
Phenylalanine	28.40 ± 1.04	35.78 ± 2.06	32.44 ± 1.25	39.50 ± 1.53	1.134 *	1.391 *	0.944	1.103
Bilirubin	3.63 ± 0.23	3.28 ± 0.45	1.85 ± 0.43	2.38 ± 0.33	0.476 *	0.656 *	0.561 *	0.726
Edetic acid	538.88 ± 19.47	331.87 ± 38.43	406.57 ± 36.52	337.50 ± 28.51	0.773 *	0.626 *	1.109	1.059
PE 38:4	1.79 ± 0.53	7.85 ± 1.06	4.35 ± 0.91	7.99 ± 0.78	2.075 *	4.459 *	0.388 *	1.018
PC 34:1	256.56 ± 8.13	349.48 ± 16.05	325.19 ± 18.49	356.50 ± 11.91	1.134	1.389 *	0.914	1.020
PC 32:0	9.42 ± 0.72	20.09 ± 1.41	13.04 ± 1.29	18.37 ± 1.05	1.191	1.949 *	0.578 *	0.914
PC 36:1	13.41 ± 1.74	28.32 ± 3.42	16.74 ± 3.51	25.66 ± 2.54	0.966	1.915 *	0.583 *	0.906
PC 38:6	40.26 ± 2.38	25.86± 4.71	22.82 ± 4.44	14.37 ± 3.49	0.543 *	0.357 *	0.806	0.558 *
PC 36:4	84.35 ± 6.46	77.60 ± 12.76	27.48 ± 2.44	20.19 ± 9.46	0.358 *	0.239 *	0.390 *	0.226 *
PC 38:3	19.03 ± 2.93	30.59 ± 5.58	12.72 ± 4.42	34.77 ± 4.14	0.546	1.827 *	0.405 *	1.136
LysoPC 16:0	37.22 ± 0.85	31.11 ± 1.69	32.24 ± 1.36	29.18 ± 1.25	0.884 *	0.784 *	1.072	0.938
LysoPC 20:4	62.53 ± 2.73	43.35 ± 5.39	38.15 ± 4.38	46.91 ± 4.00	0.638 *	0.750 *	0.889	1.082
LysoPC 18:2	346.08 ± 9.54	227.07 ± 18.84	212.09 ± 16.21	172.17 ± 13.97	0.636 *	0.497 *	0.965	0.758 *

Abbreviation: T2DM, type 2 diabetes mellitus; DKD, diabetic kidney disease; PC, phosphatidylcholine; PE, phosphatidylethanolamine; LysoPC, lysophosphatidylcholine. ^a^ fold change was adjusted for age, sex, body mass index (BMI), and chronic diseases (hypertension and hyperlipidemia). ^#^ fold change and *p* value were calculated using two sample *t* tests. * *p* < 0.05.

**Table 4 diagnostics-12-02626-t004:** Association of the metabolite combinations in the CSF samples from patients with diabetic kidney disease with current DKD or new-onset DKD.

Significantly Changed Metabolites in CSF	AIC ^#^	AUC	Adjusted OR ^a,#^	AIC ^#^	AUC	Adjusted OR ^a,#^
Correlating with current DKD	Control vs. Current DKD			Without DKD vs. Current DKD		
Paraxanthine	98.80	0.73	0.919 *	68.52	0.63	0.990
Uric acid	89.26	0.84	1.048 *	60.33	0.74	1.026
Paraxanthine, uric acid	85.27	0.85	2.718 *	62.31	0.75	2.718 *
Correlating with new-onset DKD	Control vs. New-onset DKD			Without DKD vs. New-onset DKD		
Tryptophan	80.19	0.86	2.718 *	73.46	0.745	1.018 *
Creatinine	101.32	0.85	0.559 *	82.06	0.57	0.772 *
Hypoxanthine	127.36	0.71	0.867 *	78.34	0.63	0.949
Creatinine, Tryptophan	58.68	0.95	2.718 *	74.47	0.74	2.718 *
Creatinine, hypoxanthine, tryptophan	55.52	0.96	2.718 *	70.47	0.79	2.718 *

Abbreviation: AIC, Akaike Information Criterion; AUC, area under receiver-operative-character curve; OR, odds ratio; DKD, diabetic kidney disease. ^a^ Adjusted for sex, body mass index (BMI), and chronic diseases (hypertension and hyperlipidemia). ^#^ AIC value and odds ratio were calculated using logistic regression model. * *p* < 0.05.

**Table 5 diagnostics-12-02626-t005:** Association of metabolite combinations in plasma samples from patients with diabetic kidney disease with current DKD or new-onset DKD.

Significantly Changed Metabolites in Plasma	AIC ^#^	AUC	Adjusted OR ^a,#^	AIC^#^	AUC	Adjusted OR ^a,#^
Correlating with current DKD	Control vs. Current DKD			Without DKD vs. Current DKD		
Bilirubin	101.95	0.72	0.605 *	64.54	0.65	0.663 *
PC 36:4	90.03	0.77	0.823 *	63.87	0.66	0.885 *
PE 38:4	83.71	0.82	1.502 *	62.67	0.69	0.879 *
Uric acid	56.72	0.93	1.366 *	50.09	0.88	1.241 *
L-acetylcarnitine	56.72	0.93	1.237 *	64.59	0.77	1.057 *
L-Acetylcarnitine, Uric acid	34.73	0.98	2.718 *	51.68	0.88	2.718 *
L-Acetylcarnitine, Uric acid, PC 36:4	36.32	0.98	2.718 *	52.30	0.87	2.717 *
L-Acetylcarnitine, uric acid, PC 36:4, PE 38:4	11.62	0.99	2.718 *	48.26	0.90	2.718 *
Correlating with new-onset DKD	Control vs. New-onset DKD			Without DKD vs. New-onset DKD		
LysoPC 18:2	56.39	0.95	0.9681 *	83.18	0.59	0.993 *
Uric acid	88.51	0.92	1.3038 *	78.27	0.63	1.138 *
PC 36:4	92.03	0.88	0.9641 *	72.05	0.72	0.969 *
PC 38:6	101.38	0.85	0.9234 *	80.29	0.65	0.967 *
Uric acid, PC 38:6	22.26	0.99	2.718 *	75.30	0.72	2.718 *
Uric acid, PC 36:4	75.68	0.94	2.718 *	64.61	0.82	2.718 *
Uric acid, PC 38:6, PC 36:4	23.96	0.99	2.718	68.47	0.82	2.718 *

Abbreviation: AIC, Akaike information criterion; AUC, area under receiver-operative-character curve; OR, odds ratio; DKD, diabetic kidney disease; PC, phosphatidylicholine; PE, phosphatidylethanolamine; LysoPC, lysophosphatidylcholine. ^a^ Adjusted for sex, body mass index (BMI), and chronic diseases (hypertension and hyperlipidemia). ^#^ AIC value and odds ratio were calculated using the logistic regression model. * *p* < 0.05.

**Table 6 diagnostics-12-02626-t006:** Correlation of the log2 transformed CSF and plasma metabolites with the renal function measurements in patients with T2DM.

Renal Function Measurement	Urinary Albumin/Creatinine Ratio		eGFR	
Correlation with Log2 Transformed Metabolites	Adjusted *r* ^a,#^		Adjusted *r* ^a,#^	
Significantly changed metabolites in CSF	2017	2021	2017	2021
Tryptophan	−0.132	−0.083	−0.075	−0.086
D-glucose	−0.136	−0.407 *	0.084	0.343 *
Uric acid	0.316 *	0.353 *	−0.244 *	−0.076
Paraxanthine	−0.228 *	−0.309 *	0.363 *	0.051
Hypoxanthine	0.252 *	0.381 *	−0.403 *	−0.135
Creatinine	0.021	−0.073	0.343 *	0.313 *
Significantly changed metabolites in plasma	**2017 **	**2021 **	**2017 **	**2021 **
Uric acid	0.232 *	0.306 *	−0.375 *	−0.302 *
L-Acetylcarnitine	0.186	0.514 *	−0.513 *	−0.221 *
Bilirubin	−0.073	0.029	0.170	0.375 *-
LysoPC 18:2	0.159	−0.144	0.239 *	0.299
PC 38:6	−0.136	−0.109	0.122	0.229 *
PC 36:4	−0.321 *	−0.250 *	0.253 *	0.042
PE 38:4	−0.429 *	−0.344 *	0.300 *	0.279 *

Abbreviation: CSF, cerebrospinal fluid; eGFR, estimated glomerular filtration rate; PC, phosphatidylicholine; PE, phosphatidylethanolamine; LysoPC, lysophosphatidylcholine. ^a^ Adjusted for age, sex, body mass index (BMI), and chronic diseases (hypertension and hyperlipidemia). ^#^
*r* was calculated using the regression model. * *p* < 0.05.

## Data Availability

The raw data could be available by contacting the corresponding author.

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
