# Peer review of "Metabolomic Signature of Diabetic Kidney Disease in Cerebrospinal Fluid and Plasma of Patients with Type 2 Diabetes Using Liquid Chromatography-Mass Spectrometry"

_diagnostics, 2022, doi:10.3390/diagnostics12112626_

Round 1

Reviewer 1 Report

This paper describes an untargeted metabolomics approach to discover metabolomic signature of DKD in CSF and plasma of patients with T2DM. The research subject is interesting and promising. Therefore, the reviewer thinks that this paper would be acceptable to publish on “Diagnostics”. The following points would be helpful to improve this manuscript.

1. Second paragraph in Introduction section included some redundant explanation. The reviewer thinks that, in Introduction section, second paragraph could be more condensed and combined with first or third paragraph in Introduction section.

2. In section 2.2. . CSF and blood sampling procedures, please delete full stop(.) on subheading.

3. In section 2.2. CSF and blood sampling procedures, please cite reference for first sentence.

4. In section 3.1. group separation and their demographic comparison, please use capital letter for first word on subheading.

Author Response

Comment(s) from reviewer #1:
This paper describes an untargeted metabolomics approach to discover metabolomic signature of DKD in CSF and plasma of patients with T2DM. The research subject is interesting and promising. Therefore, the reviewer thinks that this paper would be acceptable to publish on “Diagnostics”. The following points would be helpful to improve this manuscript.

Point 1: Second paragraph in Introduction section included some redundant explanation. The reviewer thinks that, in Introduction section, second paragraph could be more condensed and combined with first or third paragraph in Introduction section.

Response: We thank the reviewer for kind comment. We have condensed the Introduction section by combined the second and third paragraph and that read” The kidneys are metabolically active during the modulation of circulating metabolites and waste removal via filtration, reabsorption, and secretion. Chronic hyperglycemia and inflammation shift the oxidative balance toward a pro-oxidative state, accelerating kidney damage and causing DKD progression from glomerular hyperfiltration to declining glomerular filtration, progressive albuminuria, and ultimately ESRD [6,7]. Hence, metabolic and epigenetic dysregulation may be detected in blood and urine before the onset and diagnosis of DKD [7].”(please see Introduction section, page 2, line 60-64)

Point 2: In section 2.2. CSF and blood sampling procedures, please delete full stop(.) on subheading.

Response: We thank the reviewer for kind comment. We have deleted full stop(.) on subheading of section 2.2. (please see Materials and Methods section, page 3, line 186)

Point 3: In section 2.2. CSF and blood sampling procedures, please cite reference for first sentence.

Response: We thank the reviewer for kind comment. We have added reference for the first sentence and that read “The sample collection procedures were largely the same as those used in our previous NMR metabolomic study of patients with T2DM [13].”(please see Materials and Methods section, page 3, line 187-188)

  1. Lin, H.T.; Cheng, M.L.; Lo, C.J.; Lin, G.; Lin, S.F.; Yeh, J.T.; Ho, H.Y.; Lin, J.R.; Liu, F.C. (1)H Nuclear Magnetic Resonance (NMR)-Based Cerebrospinal Fluid and Plasma Metabolomic Analysis in Type 2 Diabetic Patients and Risk Prediction for Diabetic Microangiopathy. J Clin Med 2019, 8, doi:10.3390/jcm8060874.

Point 4: In section 3.1. group separation and their demographic comparison, please use capital letter for first word on subheading.

Response: We thank the reviewer for kind comment. We have revised the manuscript and that read “3.1. Group separation and their demographic comparison ”(please see Result section, page 4, line 260)

Reviewer 2 Report

The manuscript entitled “ Metabolomic signature of diabetic kidney disease in cerebro- 2 spinal fluid and plasma of patients with type 2 diabetes using 3 liquid chromatography-mass spectrometry” The manuscript is organized and has a merit, organized and well presented.  I think the authors should improve the presentation of their aim and hypothesis to introduce more clear idea for the readers. There is also a major concern regarding about the hypothetical base : based on your aim : the metabolomic signatures of patients with DKD in CSF samples have not been well explored. To profile the causal relationship of these metabolic 90 alterations in such patients who develop DKD, we conducted this longitudinal follow-up 91 study of CSF and plasma samples from patients with T2DM and DKD in comparison with 92 patients with T2DM but without DKD and control participants using sensitive LC-MS 93 analysis.

Why did the authors decide to correlate the changes in CSF metabolimic with the CKD in T2DM what is the vision and proper conclusion for the clinical applications, Why not to concentrate on the biogenic markers in plasma , serum or urine which are accessible candidates to estimate KD easily and with our the painful methods for CSF collection.

-        Is there an ethical consent code for the study , were the patients included?

-        Who applied the CSP sampling and anesthesia for patients.

-        Line 54 :  plz cite this relevant paper “10.1016/j.lfs.2021.119674

-        OPLS which is on the subtitles should be written complete

-        Please, add the sufficient citations in the methods section for Biochemical analyses of CSF and plasma samples.

-        The model ,and country of LC-MS analysis.

-        The % of change between the control and other conditions should be estimated in the results text to spot light on the pronounced changes.

Round 2

Reviewer 2 Report

The manuscript is now improved. The authors addressed the reviewer comments well and I found it is now suitable for publication after modification of the conclusion to be in accordance with the aim of the work 

Author Response

We thank the reviewer for kind comment. We have refined our conclusion to be in accordance with the aim of the work and that read “In this longitudinal study, we profiled the CSF and plasma metabolomic signature of patients with T2DM complicated with current DKD or developing new-onset DKD using LC-MS analysis. The identified potential DKD biomarkers, such as uric acid, acetylcarnitine, PC 36:4, and PE 38:4, had significant correlation with eGFR and UACR, suggesting their values in predictions and risk stratification for DKD in patients with T2DM. These metabolic alterations imply defective mitochondrial fatty acid oxidation, purine metabolism, and phospholipid remodeling during DKD progression. Further verification of our results in a larger multi-omic cohort is required to confirm the causal relationship.”(please see Conclusions section, page 15, line 549-557)